# Study on the China's real interest rate after including housing price factor into CPI

**Shiting Ding**[ID], **Qintian Pan, Yanming Zhang, Jingru Zhang, Qiong Yang, Jingdong Luan**[ID]*

School of Economics and Management, Anhui Agricultural University, Hefei, Anhui, P. R. China

* luanjingdong@ahau.edu.cn

**Data Availability Statement:** The data source were revised as follow in footnote 1 with the data link: Data source: "China Real Estate Statistical Yearbook" "China Statistical Yearbook" "China Financial Statistical Yearbook", and all of those

## Abstract

The Chinese economy has undergone a long-term transition reform, but there is still a planned economy characteristic in the financial sector, which is financial repression. Due to the existence of financial repression, China's actual interest rate level should be lower than the Consumer Price Index (CPI). However, based on official China's interest rates and CPI, over half of the years China's actual interest rate remained higher than CPI by our calculation from 1999 to 2022. This is inconsistent with the financial repression that exists in China, and the main reason is the calculation methods of China's CPI. China's CPI measurement system originated from the planned economy era, which did not fully consider the rise in housing purchase prices, so the current CPI measurement system can be more realistically presented by taking the rise in housing prices into consider. The core idea of this study is to mining relevant official statistical data and calculate the proportion of Chinese residents' expenditure on purchasing houses to their total expenditure. By taking the proportion of house purchases as the weight of house price factor, and taking the proportion of other consumption as the weight of official CPI, the Generalized CPI (GCPI) is formulated. The GCPI is then compared with market interest rates to determine the actual interest rate situation in China over the past 20 years. This study has found that if GCPI is used as a measure, China's real interest rates have been negative for most years since 1999. Chinese residents have suffered the negative effects of financial repression over the past 20 years, and their property income cannot keep up with the actual losses caused by inflation. Therefore, it is believed that China's CPI calculation method should be adjusted to take into account the rise in housing prices, so China's actual inflation level could be more accurately reflected. In view of the above, deepening interest rate marketization reform and expand channels for financial investment are the future development goals of China's financial system.

## 1. Introduction

In many developing countries, due to the existence of financial repression [1,2], the interest rates in their banking systems have been artificially lowered [3], and because of the shortcomings of inflation measurement methods [4], their real interest rates has always been a "black

yearbooks can be found in "China Statistical Yearbooks Database" with the link: https://data.cnki.net/.

**Funding:** The author(s) received no specific funding for this work.

**Competing interests:** The authors have declared that no competing interests exist.

box" that is difficult to know [5]. China is the largest developing country in the world, and in order to provide low-cost funding for the industrial sector, she has also implemented a long-term policy of financial repression since planned economy era [6].

China's real interest rate is a hot research topic in finance [7–9]. From the perspective of theoretical analysis, financial repression often leads to negative real interest rates [10]. Therefore, China's long-term financial repression policies should lead to negative real interest rates [11], and some studies have pointed out that if China further optimizes its interest rate management system, it will probably better contribute to economic development and provide a model for other countries [12,13]. It can be seen that accurate evaluation of the real interest rate level of China's financial system has positive implications for policy implementation and academic research.

The academic community has adopted different methods to calculate China's real interest rate, but has not reached completely consistent conclusions. Huang et al (2010) confirm that excess liquidity, output gap, housing prices, and stock prices positively affect inflation [14], and He et al (2014) believe that after China completes the marketization of interest rates, China's interest rates will rise [15]. However, after China formally completed the market-oriented reform of interest rates in October 2015, the market deposit interest rate did not significantly increase, mainly due to the window guidance of the People's Bank of China on bank deposit interest rates [16], which means that the financial repression in China's financial system has not been completely eliminated [17]. Measuring China's real interest rate level remains a research question worthy of attention [18].

Financial repression often lead to negative real interest rates [19]. There are several classical methods to estimate the real interest rates [20,21], and the main method of calculating the real interest rate of residents is to deduct the nominal deposit interest rate from the inflation rate [22]. The commonly used indicator to measure inflation is CPI. However, the CPI used by China's official statistics department originates from the calculation method of the planned economy era, and cannot fully effectively evaluate the actual inflation level of the Chinese economy [23]. Classical financial theory believes that the actual interest rate should be equal to the nominal interest rate minus the inflation rate [24]. In China, the nominal interest rate is described by the one-year fixed deposit rate [25], while the inflation rate is described by the CPI of the statistical department [26]. Unfortunately, using the above method to measure China's real interest rate is not accurate, because on the one hand, the fixed deposit interest rate cannot timely reflect the changes in interest rate market [27], and on the other hand, the CPI of the China Bureau of Statistics cannot fully represent China's real inflation level [28].

In response to the problem that market interest rates cannot be fully represented by one-year fixed deposits, academia has pointed out that the Shanghai Interbank Offered Rate (SIBOR) can be used as a measurement indicator of market interest rates [29], while many studies have pointed out that house consumer price factors should be taken into account in the revision of CPI [30], since the housing prices are rising too fast and costed a lot [31]. China's CPI does indeed include changes in housing rent, but the rent-price ratio of Chinese housing is lower than the widely recognized level for longtime [32], and purchasing house in China is a necessity among residents [33]. Therefore, considering only housing rents cannot represent the level of expenditure by residents. This study referenced the CPI calculation methods of developed countries such as the United States that consider housing purchase factors [34], and attempted to incorporate the change in housing purchase expenditure prices into the CPI measurement system, in order to better characterize China's actual inflation rate.

The objective of this study is to revise China's official nominal interest rate and inflation rate by representing the market nominal rate with SIBOR interest rate and by considering the GCPI of housing prices, in order to get a more real level of China's real interest rate. The

specific measures include the following two points: first, using SIBOR interest rate as the measurement standard of market interest rate (nominal interest rate); second, adopting a certain method to incorporate the rise of housing prices into the existing CPI system, thereby correcting the limitations of China's existing CPI measurement. The marginal academic contribution of this study is considering the revision of China's nominal interest rate and official inflation rate at the same time, and using the revised indicators to measure China's real interest rate level in the past 20 years, thus providing empirical support for judging China's financial deepening level.

The outline of this study mainly includes: Relevant data are processed to provide a basis for further comparison within the same time range. The proportion of housing purchase expenditure is calculated in total household consumption, and use this weight to modify China's official CPI into GCPI. SHIBOR interest rate is used to replace the nominal interest rate, and use SHIBOR interest rate to minus GCPI to obtain China's real interest rate in recent years. It is found that China's real interest rate is negative in most of recent years, which is quite different from the results calculated directly with China's official data.

## 2. Methods and materials

### 2.1 Treatment of data

Due to the fact that the SIBOR is almost traded every working day, the SIBOR rate changes daily [35,36]. The People's Bank of China adjusts the fixed deposit interest rate every 1–2 years on average [37]. Overall, the China Bureau of Statistics publishes CPI data every month, and the annual statistical bulletin publishes China's annual CPI data. Although the China Bureau of Statistics publishes monthly housing price increases in 70 large and medium-sized cities, the average increase in urban housing prices across the country is based on annual data only. In order to better compare these four data, annual data on SIBOR interest rates, CPI levels, and housing price increases were selected, and they were set on June 30th of each year. The deposit level was selected at the time when the Chinese central bank actually adjusted the deposit rate. Through such processing, four sets of data can be unified into a time series with a span of years, and then China's real interest rate with a longer time scale can be studied.

### 2.2 Processing ideas for incorporating housing prices into CPI

The rise in housing prices is an important social phenomenon in China, and China's CPI include the price changes of the daily consumption objects of the residents. However, this residential price only includes construction and decoration materials, housing rent, self owned housing, water, electricity, and fuel, but does not include the largest expenditure on housing purchases. Studies have also pointed out that this is an important shortage in China's CPI measurement [38,39]. Therefore, the main marginal contribution of this study is to estimate the proportion of housing purchase expenditure per capita, and then collect data on housing price trends. By integrating the changes in housing prices with the changes in the original CPI, the generalized price changes in China after incorporating housing prices were formulated.

### 2.3 Measurement method of GCPI including housing prices

Since the current CPI of China's official statistical department does not include the increase in housing prices [40], if we can calculate the proportion of housing purchase in total expenditure, the GCPI included in house consumption can be obtained through the weighted sum

method, and the specific calculation equations are as follows:

$$HPP_t = HS_t/TP_t$$

$$(1)$$

$$HPP_t + CPP_t = PP_t$$

$$(2)$$

$$GCPI_t = CPI_t * \frac{CPP_t}{PP_t} + HPI_t * \frac{HPP_t}{PP_t}$$

$$(3)$$

In Eq 1, HPP represents the per capita house consumption amount in year t, which is equal to HS divided by TP. HS represents the total national house sale amount in year t, and TP represents the national population in year t. The data for HS and TP are from the China Real Estate Statistical Yearbook and the China Statistical Yearbook. In Eq 2, CPP represents the per capita consumption expenditure corresponding to the official CPI in year t, while PP represents the total per capita consumption expenditure in year t. CPP is the object to be calculated, and the value of PP is derived from the China Statistical Yearbook. In Eq 3, GCPI represents the Generalized CPI in year t, while HPI represents the change in housing prices in year t. GCPI is the object of calculation, and the value of HPI is sourced from the average sales prices of house properties over the years in the China Real Estate Statistical Yearbook.

## 2.4 Measurement of China's real interest rate based on GCPI

In order to evaluate China's real interest rate after deducting inflation from different perspectives [41], four indicators were selected: CPI, GCPI, one-year fixed deposit rate, and SIBOR. The traditional real interest rate and the real interest rate calculation based on GCPI are analyzed separately, and the results were compared. By subtracting the inflation rate (CPI and GCPI) from the nominal interest rate (SIBOR and one-year fixed deposit), this paper shows the calculation results of China's real interest rate from different perspectives. Different from relevant studies that focus on the influencing factors of China's real estate market price (including the impact of market interest rates on housing prices) [42,43], one of the innovations of this paper is to study the changes of China's real interest rates after housing prices are included from the data level. This study updates the calculation of China's real interest rates on one hand, and also provides reference for academic research and policy formulation on the other.

## 2.5 Presentation of relevant data

The following Table 1 shows the detail data of each variables.

# 3. Results and discussions

## 3.1 Real interest rate calculation based on official data from the Chinese government

According to the official one-year fixed deposits and CPI data of China, taking one-year fixed deposits as the nominal interest rate and CPI as the inflation rate, the following results can be obtained in Fig 1 from 1999–2022:

According to the results of the above figure, it can be found that if calculated using official statistical data released by China, from 1999 to 2022, there were 13 years when China's actual deposit interest rate was positive, which means the interest rate of one-year fixed deposits was

**Table 1. Presentation of data values.**

| Year | 1999 | 2000 | 2001 | 2002 | 2003 | 2004 | 2005 | 2006 | 2007 | 2008 | 2009 | 2010 |
|---|---|---|---|---|---|---|---|---|---|---|---|---|
| HS[1] | 125300 | 126300 | 127200 | 128000 | 128800 | 129600 | 130400 | 131100 | 131800 | 132500 | 133100 | 125300 |
| TP[2] | 124200 | 125300 | 126300 | 127200 | 128000 | 128800 | 129600 | 130400 | 131100 | 131800 | 132500 | 133100 |
| PP[3] | 4868 | 5480 | 5893 | 6473 | 6903 | 7474 | 8224 | 8713 | 9361 | 9258 | 10069 | 10919 |
| HPI[4] | 0.16% | 4.90% | 3.54% | 3.72% | 5.02% | 18.71% | 12.58% | 6.23% | 16.86% | -1.89% | 24.69% | 5.97% |
| CPI[5] | -1.40% | 0.35% | 0.72% | -0.73% | 1.13% | 3.82% | 1.78% | 1.65% | 4.82% | 5.93% | -0.73% | 3.18% |
| Year | 2011 | 2012 | 2013 | 2014 | 2015 | 2016 | 2017 | 2018 | 2019 | 2020 | 2021 | 2022 |
| HS[1] | 133800 | 134500 | 135400 | 136300 | 137200 | 138000 | 138800 | 139600 | 140300 | 140800 | 141100 | 141200 |
| TP[2] | 134500 | 135400 | 136300 | 137200 | 138000 | 138800 | 139600 | 140300 | 140800 | 141100 | 141200 | 141175 |
| PP[3] | 11613 | 12278 | 12850 | 13894 | 15589 | 17869 | 18322 | 19853 | 21559 | 21210 | 24100 | 24538 |
| HPI[4] | 5.67% | 8.75% | 7.73% | 1.42% | 9.10% | 11.28% | 5.71% | 12.33% | 8.58% | 7.46% | 4.17% | -2.03% |
| CPI[5] | 5.55% | 2.62% | 2.62% | 1.92% | 1.44% | 2.00% | 1.59% | 2.07% | 2.90% | 2.42% | 0.98% | 2.00% |

1.Ten thousand yuan

2.Ten thousand people

3.Yuan

4.Percent

5.Percent.

Data source: "China Real Estate Statistical Yearbook" "China Statistical Yearbook" "China Financial Statistical Yearbook", and the all of those Yearbook can be found in "China Statistical Yearbooks Database" with the link: https://data.cnki.net/.

higher than CPI more than half years from 1999–2022. So if we take into account the rise in housing prices, will there be any changes in the results?

## 3.2 Calculation results of GCPI

The starting point of China's official statistics department's housing price data is 1998. In order to measure the increase in housing prices, this article selects 1999 as the starting point for the rise in housing prices. Through the introduction of housing price data, the GCPI data is recalculated as follows in Table 2 and Fig 2.

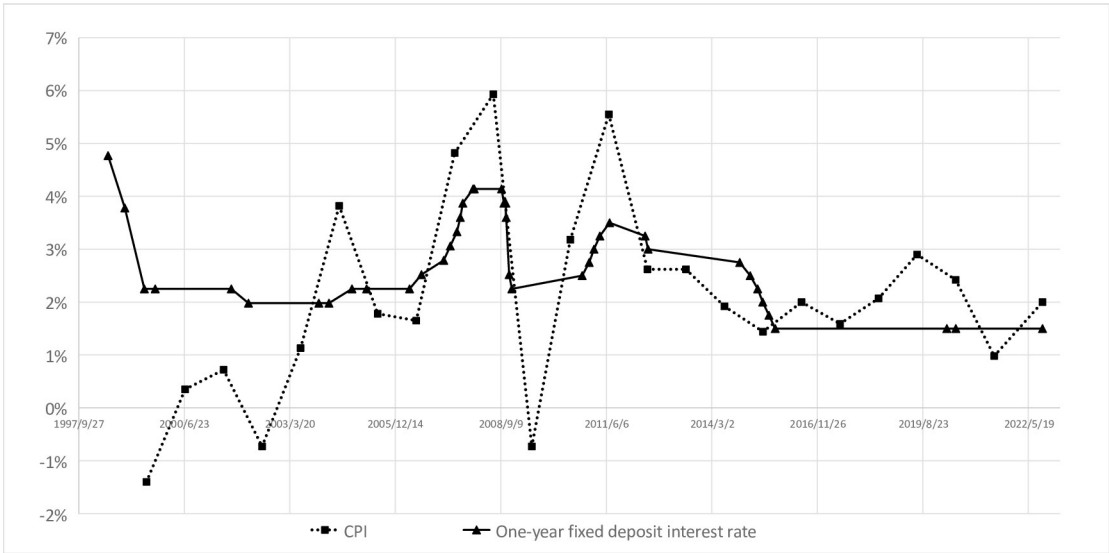

**Fig 1. Comparison of CPI and one-year fixed deposit interest rate.**

**Table 2. Comparison of CPI and GCPI.**

| Year | 1999 | 2000 | 2001 | 2002 | 2003 | 2004 | 2005 | 2006 |
|---|---|---|---|---|---|---|---|---|
| CPI | -1.40% | 0.35% | 0.72% | -0.73% | 1.13% | 3.82% | 1.78% | 1.65% |
| GCPI | -1.34% | 0.56% | 0.87% | -0.46% | 1.42% | 5.14% | 3.25% | 2.34% |
| Year | 2007 | 2008 | 2009 | 2010 | 2011 | 2012 | 2013 | 2014 |
| CPI | 4.82% | 5.93% | -0.73% | 3.18% | 5.55% | 2.62% | 2.62% | 1.92% |
| GCPI | 7.32% | 4.58% | 6.56% | 4.02% | 5.59% | 4.59% | 4.60% | 1.76% |
| Year | 2015 | 2016 | 2017 | 2018 | 2019 | 2020 | 2021 | 2022 |
| CPI | 1.44% | 2.00% | 1.59% | 2.07% | 2.90% | 2.42% | 0.98% | 2.00% |
| GCPI | 4.03% | 5.71% | 3.36% | 6.73% | 5.51% | 5.02% | 2.50% | 0.64% |

The data comes from the calculation of Eqs 1–3.

By comparing GCPI and CPI, it can be found that in most years, the GCPI is higher than the value of CPI. In the 24 years from 1999 to 2022, only in 2008, 2014, and 2022 the GCPI was lower than the CPI. Among them, 2008 was the year of the global financial crisis, 2014 was the year when the Chinese government introduced strict housing purchase restrictions, and 2022 was the year when Chinese government has launched a strict policy to prevent and control the COVID-19. In other years, due to the rapid rise in housing prices, the general CPI is significantly higher than CPI. It can be seen that the CPI of China's statistical department may have underestimated the overall inflation level of consumption.

### 3.3 Measurement of China's real interest rate based on GCPI

On the basis of the measurement of GCPI, this paper adopts two methods to calculate the actual interest rate, one is to use the one-year fixed deposit interest rate as the nominal interest rate to deduct the GCPI, the other is to use SIBOR as the nominal interest rate to deduct the

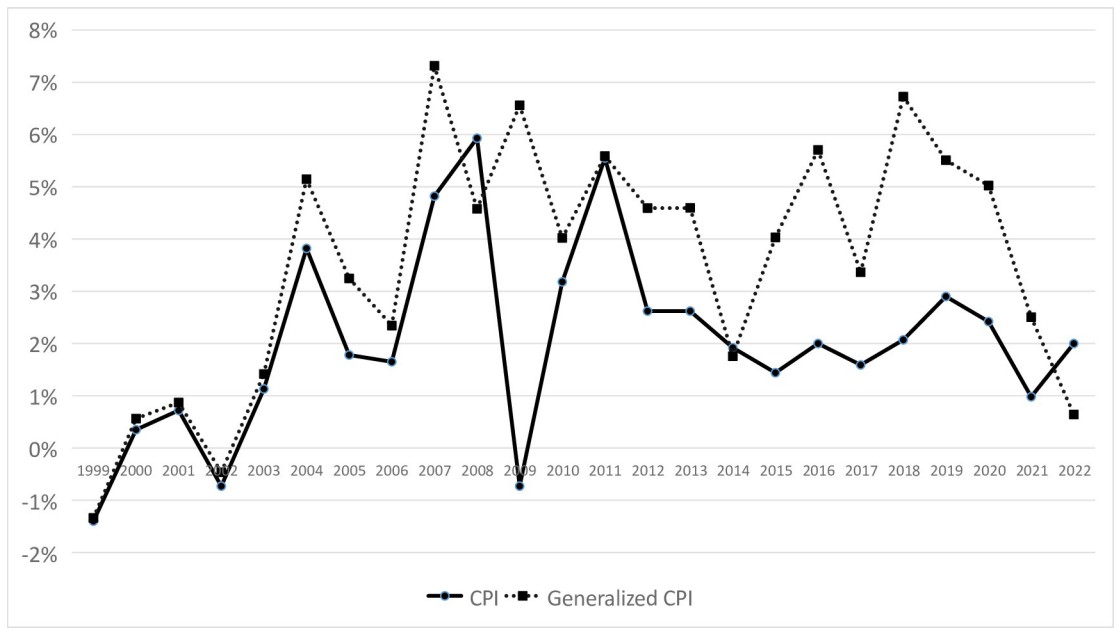

**Fig 2. Comparison of CPI and generalized CPI.**

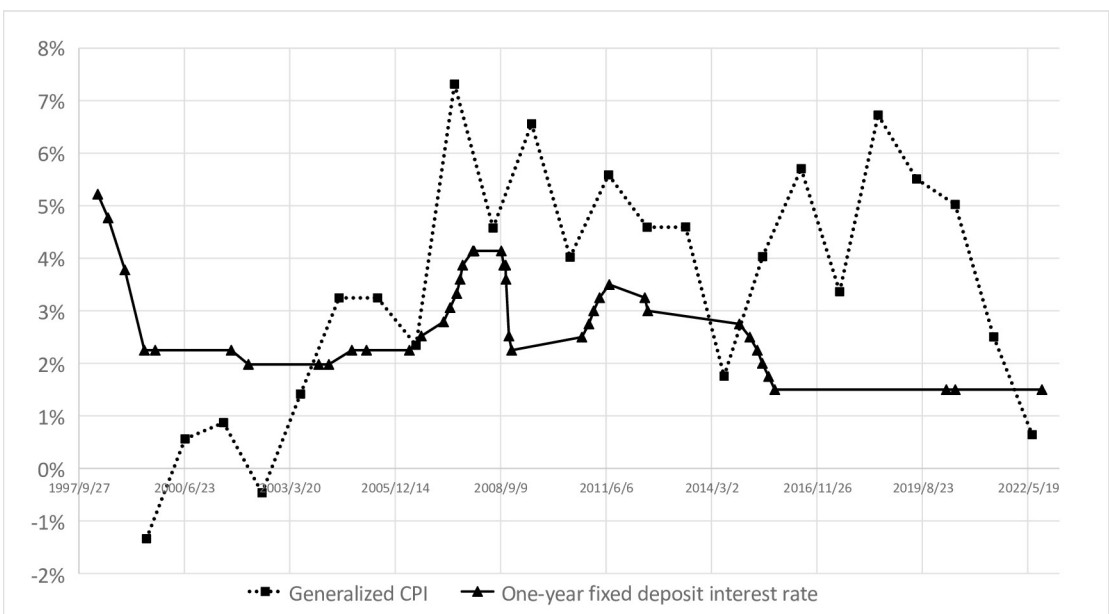

**Fig 3. Comparison of generalized CPI and one-year fixed deposit interest rate.**

GCPI. Due to the financial repression in China's financial system, the first method may underestimate China's nominal interest rate, while the second method is closer to China's real interest rate level. The specific calculation results by first method are shown as follow in Fig 3.

It can be seen from the line chart above that China's one-year fixed deposit interest rate was generally higher than the GCPI level from 1999 to 2004, but since 2004, due to the rapid rise of housing prices, China's one-year fixed deposit interest rate has been lower than the GCPI level for a long time. From this perspective, the actual return on bank deposits in China has been at a negative level for nearly 20 years. Comparing Figs 1 and 3, it can be found that if housing prices are included in CPI the years when China's real interest rate is positive have significantly decreased according to the our calculation.

However, since one-year fixed deposits cannot fully represent the interest rate level in the market [44], the following Table 3 and Fig 4 show the actual interest rate level in China after taking SIBOR as the market nominal interest rate and deducting the GCPI.

It can be seen from the above table and figure, even if SIBOR is used as the representative of nominal interest rate, the real interest rate level in China is still negative in most years since 2006 (The starting point of SIBOR). In China residents have other investment opportunities

**Table 3. Comparison of GCPI and SIBOR.**

| Year | 2006 | 2007 | 2008 | 2019 | 2010 | 2011 | 2012 | 2013 | 2014 |
|------|------|------|------|------|------|------|------|------|------|
| GCPI | 2.34% | 7.32% | 4.58% | 6.56% | 4.02% | 5.59% | 4.59% | 4.60% | 1.76% |
| SIBOR | 2.43% | 2.08% | 2.33% | 1.01% | 1.62% | 3.28% | 2.85% | 3.28% | 2.79% |
| Year | 2015 | 2016 | 2017 | 2018 | 2019 | 2020 | 2021 | 2022 | |
| GCPI | 4.03% | 5.71% | 3.36% | 6.73% | 5.51% | 5.02% | 2.50% | 0.64% | |
| SIBOR | 2.06% | 2.06% | 2.61% | 2.49% | 2.21% | 1.62% | 1.93% | 1.49% | |

The SIBOR interest rate data was first released in 2006, so 2006 is used as the starting year for comparison. Using the daily interest rate of SIBOR on June 30th each year as a representative of the annual SIBOR.

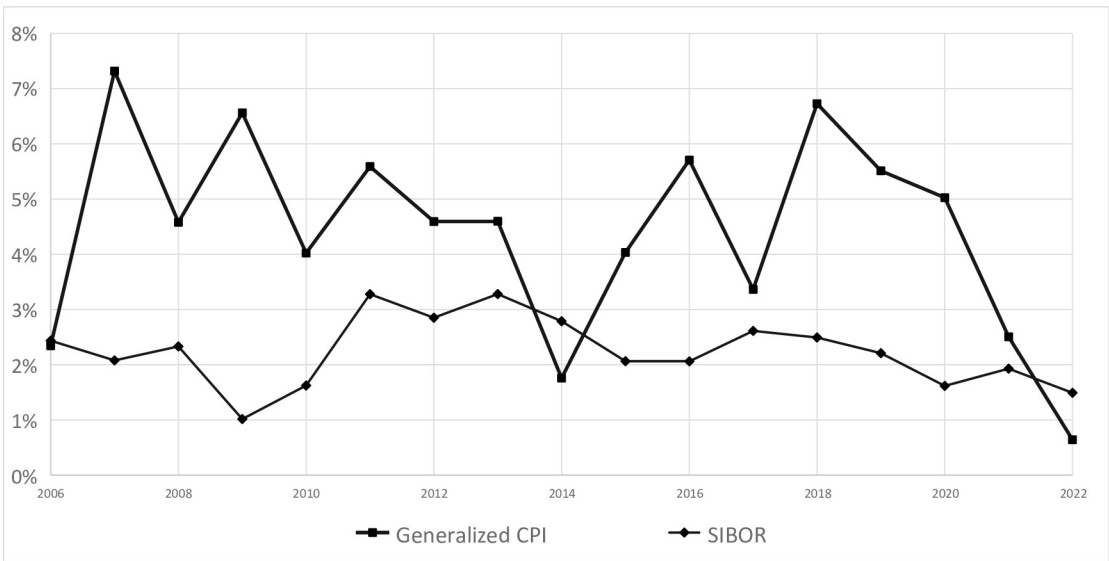

**Fig 4. Comparison of generalized CPI and SIBOR.**

with lower risks, such as monetary funds, but the returns of monetary funds generally lower than SIBOR. From this, it can be seen that Chinese residents still bear the long-term negative real interest rate level take housing price into consider.

### 3.4 What changes have GCPI brought to China's real interest rate calculation

Both the ordinary people and relevant scholars believe that China's actual inflation level is underestimated by the official CPI [45], and the most crucial reason is that the official CPI does not include the rapid rise and high cost of housing purchase. This study is based on public statistical data and uses a clear weighting methods to synthesize the weights of residents' house purchases increase in house prices with official CPI, generating a new GCPI. In the past 24 years, if GCPI is used as the representative of inflation, whether SIBOR or one-year fixed deposit is used as the nominal interest rate, China's real interest rate in most years is negative. This result is not only in line with the perception of the ordinary people, but also reflected the the financial repression policies that still exist in China. It is worth noting that in the second decade of the 21st century, due to the gradual cooling of China's real estate market, the actual value of GCPI showed a lower trend than CPI. This result confirms the accuracy of using GCPI as an inflation measure from another perspective, as research has pointed out that there is a risk of deflation in the current Chinese economy [46].

### 4. Conclusions and implications

From a theoretical perspective, China's long-term implementation of financial repression policies should result in negative real interest rates. However, based on official Chinese interest rates and CPI, real interest rates are positive in 13 years of 1999 to 2022. According to the research in this paper, the main reason for this phenomenon is that the official CPI measurement from China during the planned economy era underestimated the actual price increase. This paper selects statistical data from 1999 to 2022 and incorporates the rise in housing prices during this period into the CPI data using a weighted method, obtaining a GCPI. The study

finds that if the GCPI is used as a measurement tool, whether the one-year fixed deposit interest rate or SIBOR represents the nominal interest rate, China's real interest rate has been negative in most years since 1999 and especially after 2004. Based on such results, the following implications can be drawn: China's CPI calculation method should be adjusted to consider more factors, especially changes in housing prices; China should continue to deepen the market-oriented reform of interest rates especially deposit interest rates; China should continue to expand channels for financial investment and provide residents with more stable investment opportunities.

The core idea of this study's calculation method is to assume that the expenditure corresponding to China's official CPI is the total expenditure of residents minus the expenditure on purchasing houses. Then, the weights of CPI and HPI are calculated through relevant statistical data, and the two are combined into GCPI. The foundation of this approach is still based on statistical data measurement, and in the future, improvements can be made from two aspects: firstly, improving CPI measurement through big data methods; another approach is to use higher-level methods such as dynamic factor analysis to improve the synthesis of HPI and CPI [47].

## Supporting information

**S1 Data.**
(XLS)

## Author Contributions

**Conceptualization:** Shiting Ding, Jingdong Luan.

**Data curation:** Qintian Pan.

**Formal analysis:** Shiting Ding, Yanming Zhang.

**Methodology:** Shiting Ding, Jingru Zhang.

**Resources:** Qiong Yang.

**Writing – original draft:** Shiting Ding.

**Writing – review & editing:** Shiting Ding.

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
