## [Decision Letter · Decision Letter 0]

16 May 2023

PONE-D-23-10280Study on the Measurement of China's Real Interest Rate after Including Housing Price FactorPLOS ONE

Dear Dr. Luan,

Thank you for submitting your manuscript to PLOS ONE. After careful consideration, we feel that it has merit but does not fully meet PLOS ONE’s publication criteria as it currently stands. Therefore, we invite you to submit a revised version of the manuscript that addresses the points raised during the review process.

We look forward to receiving your revised manuscript.

Kind regards,

Yuantao Xie

Academic Editor

PLOS ONE

Journal Requirements:

Additional Editor Comments:

Please update your manuscript according to the reviewer report.

Reviewers' comments:

Reviewer's Responses to Questions

**Comments to the Author**

1. Is the manuscript technically sound, and do the data support the conclusions?

Reviewer #1: Partly

Reviewer #2: Yes

Reviewer #3: Yes

Reviewer #4: Yes

Reviewer #5: Partly

2. Has the statistical analysis been performed appropriately and rigorously? 

Reviewer #1: No

Reviewer #2: N/A

Reviewer #3: No

Reviewer #4: Yes

Reviewer #5: I Don't Know

3. Have the authors made all data underlying the findings in their manuscript fully available?

Reviewer #1: Yes

Reviewer #2: Yes

Reviewer #3: Yes

Reviewer #4: Yes

Reviewer #5: Yes

4. Is the manuscript presented in an intelligible fashion and written in standard English?

Reviewer #1: Yes

Reviewer #2: Yes

Reviewer #3: No

Reviewer #4: Yes

Reviewer #5: Yes

5. Review Comments to the Author

Reviewer #1: Title, Study on the Measurement of China's Real Interest Rate after Including Housing Price

Factor, why do we need to include housing price factor? Interest rate is a standalone issue. China is a planned economy, meaning that whatsoever housing price is sort of under "control" of the administrative measure " As China is a planned economy country, administrative factors are one main factor that affects the housing price." See https://www.researchgate.net/publication/359519947_Factors_Affect_the_Housing_Prices_in_China_A_Systematic_Review_of_Papers_Indexed_in_Chinese_Science_Citation_Database.

Abstract part has to shorten the Background of the research, Highlight research gap that it has filled, state the point of originality, academic, practical and policy contributions of this paper.

Introduction, tells the research question, objectives and research gap that it fills.

Introduce the various section in last part.

2.1, 2.3, 2.4, missing citation

Detailed description of the data is needed.

Extend section 2.4

Interest rate has been studied extensively in previous literature of housing price research such as:

https://www.mdpi.com/2071-1050/10/2/341

https://www.researchgate.net/publication/359519947_Factors_Affect_the_Housing_Prices_in_China_A_Systematic_Review_of_Papers_Indexed_in_Chinese_Science_Citation_Database

What is the new point in this research then? Please compare your research and then with previous research to highlight the research gap that it fills. One subsection is needed for interest rate research impact on housing price.

Extend the results part and then make discussion an independent section. Discussion has to be done based on previous literature.

Calculation of interest rate CPI is not needed much. There is not much academic and practical value in new calculations of CPI and interest rate.

All tables need citations, so as the calculations.

A more complicated model is needed. Now is a bit too simple.

Extend conclusion, state limitations and future research direction.

References [J] is not needed.

Font size of the words need to be standardized.

Reviewer #2: Review of "Study on the Measurement of China's Real Interest Rate after Including Housing Price Factor" by Shiting et al. (2023)

Authors proposed a new measure of China's real interest rate that generalizes the usual Consumer Price Index (CPI). They compares CPI with proposed Generalized CPI and provide their benefits. Then they related the Generalized CPI with Shanghai Interbank Offered Rate (SIBOR) and study its impact.

I think that authors provide a new insight in the area and discuss appropriately their benefits. The new measure could be considered in future works. I could recommend the publication of this manuscript in Plos One if authors addressed the following issues:

1. Line 16: remove "generally".

2. Line 17: "CPI" <-> "Consumer Price Index (CPI)".

3. Line 80: "Second" <-> "second".

4. Line 84: "2.1. Treatment of data".

5. Eqs. (1)-(3): remove these points.

6. Replace "formula" by "Equation".

7. Line 138: "can be obtained in Figure 1 from 1999-2022.".

8. Line 222: Add as further comment the use of statistical method for the study of comparisons between Generalized CPI and SIBOR, or another important variables related to resources, such as in Idrovo-Aguirre and Contreras-Reyes (2021).

Reference: Idrovo-Aguirre, B.J., Contreras-Reyes, J.E. (2021). The response of housing construction to a copper price shock in Chile (2009–2020). Economies, 9(3), 98.

Reviewer #3: First of all, the English writing of this manuscript needs to be improved.

Secondly, it is suggested that the author add a literature review to specifically summarize the connotation and measurement methods of Real Interest Rate, and find that it is not enough to highlight the value and innovation of this study.

Finally, China's CPI includes rents, not housing prices. Usually, rents and housing prices move in tandem. Why doesn't China's CPI include housing prices? Housing is an investment property, so it is likely that housing prices rise and price of commodities remains stable or price of commodities rises and housing prices remain stable. The authors need to clarify the underlying reasons why China's CPI excludes housing prices.

Reviewer #4: I have carefully studied Manuscript Study on the Measurement of China's Real Interest Rate after Including Housing Price Factor and have the following observations for the author(s):

Q1) The abstract is too short and does not provide useful details to the readers. The authors have done a great job but they have not explained their work in the Abstract. Please follow some previous papers and provide background, findings, methodology, and brief policies in this section to make it more attractive to readers

Q2 The authors need to more empirical tests

Q3 Please, provide those theoretical reasoning that underpin the direction of the authors research investigations and the econometric model used in the study

Q4 The interpretations of results should be modified to have scientific reasoning. The author(s) need to provide the interpretations of results for each variable analysed in terms of real time happening of events. The interpretations should as well be expanded to provide the implications of both positive and negatives findings to the individual case variable under focus and to also provide the how and why the findings in the result went the way it did with respect to the case study area. Each findings discovered by the study should be compared with past studies and draw lesson and head-ways from the perspective of the study’s objectives and the relevant theories provided in the study.

Q5 Some recent literature relating to interest rate must be added. Some suggestions are as follows:

Xu, Y., Ma, Y., Zhu, Z., Li, J., & Lu, T. (2022). Construct comprehensive indicators through a signal extraction approach for predicting housing price crises. Plos one, 17(8), e0272213.

Samour, A., Isiksal, A. Z., & Gunsel Resatoglu, N. (2020). The impact of external sovereign debt and the transmission effect of the US interest rate on Turkey’s equity market. The Journal of International Trade & Economic Development, 29(3), 319-333.

Reviewer #5: - Manuscript is not well written in English

-Title can be improved ( title not catchy/ convincing at all). title is misleading, this is not a study in general, this is more like new calculation on CPI

- The first 2 sentences in the abstract does not attract the reader to read the whole paper. Can be improved further

- No literature review section/ not enough literature to support the objective of the study.

-No analysis conducted, it is just calculation of new CPI

6. PLOS authors have the option to publish the peer review history of their article (what does this mean?). If published, this will include your full peer review and any attached files.

Reviewer #1: No

Reviewer #2: No

Reviewer #3: No

Reviewer #4: No

Reviewer #5: No

---

## [Author Response · Author response to Decision Letter 0]

14 Jun 2023

Dear Yuantao and respectively reviewers:

Thank you so much for giving us the opportunity to revise and resubmit our manuscript. We are grateful to the reviewers for some constructive comments on our original paper and their support of our manuscript “Study on the Measurement of China's Real Interest Rate after Including Housing Price Factor” (Paper ID #: PONE-D-23-10280). According to your decision and the comments given by the reviewers, we have revised our manuscript carefully and highlighted the revised parts by “Track Changes” function.

For reviewer1:

Comment 1: Title, Study on the Measurement of China's Real Interest Rate after Including Housing Price. Factor, why do we need to include housing price factor? Interest rate is a standalone issue. China is a planned economy, meaning that whatsoever housing price is sort of under "control" of the administrative measure" As China is a planned economy country, administrative factors are one main factor that affects the housing price." See https://www.researchgate.net/publication/359519947_Factors_Affect_the_Housing_Prices_in_China_A_Systematic_Review_of_Papers_Indexed_in_Chinese_Science_Citation_Database.

Reply 1: Thank you for the comments. We agree that China's economic system still contains many elements of a planned economy, and we have carefully read this paper: "Factors Affect the Housing Prices in China: A Systematic Review of Papers Indexed in Chinese Science Citation Database". We also agree with the main points of this paper. However, the progress of China's interest rate market has still achieved some results. We believe that interest rate levels are an standalone issue, but the main purpose of this article is not to study the influencing factors of Chinese interest rates, but to better measure China's actual interest rate level. The real interest rate is equal to the nominal interest rate minus the inflation level, which is the most basic method for measuring the real interest rate. Because the CPI deviates from the real level due to the planned composition of China's economy as you mentioned, so we take the housing price level into consideration through certain methods to generate a generalized CPI.

Comment 2: Abstract part has to shorten the Background of the research, Highlight research gap that it has filled, state the point of originality, academic, practical and policy contributions of this paper.

Reply 2: Thank you for the comments, we have rewritten the abstract part as follow: The Chinese economy has undergone a long-term transition reform, but there is still a planned economy characteristic in the financial sector, which is financial repression. Due to the existence of financial repression, China's actual interest rate level should be lower than the Consumer Price Index (CPI). However, based on official China’s interest rates and CPI, over half of the years China’s actual interest rate remained higher than CPI by our calculation from 1999 to 2022. This is inconsistent with the financial repression that exists in China, and the main reason is the calculation methods of China's CPI. China's CPI measurement system originated from the planned economy era, which did not fully consider the rise in housing purchase prices, so the current CPI measurement system can be more realistically presented by taking the rise in housing prices into consider. The core idea of this study is to mining relevant official statistical data and calculate the proportion of Chinese residents' expenditure on purchasing houses to their total expenditure. By taking the proportion of house purchases as the weight of house price factor, and taking the proportion of other consumption as the weight of official CPI, the Generalized CPI (GCPI) is formulated. The GCPI is then compared with market interest rates to determine the actual interest rate situation in China over the past 20 years. This study has found that if GCPI is used as a measure, China's real interest rates have been negative for most years since 1999. Chinese residents have suffered the negative effects of financial repression over the past 20 years, and their property income cannot keep up with the actual losses caused by inflation. Therefore, it is believed that China's CPI calculation method should be adjusted to take into account the rise in housing prices, so China's actual inflation level could be more accurately reflected. In view of the above, deepening interest rate marketization reform and expand channels for financial investment are the future development goals of China's financial system.

Comment 3: Introduction, tells the research question, objectives and research gap that it fills.

Reply 3: Thank you for the comments, we have carefully examined the introduction part, and added the research question, objectives and research gap filled.

Research question: Measuring China's real interest rate level remains a research question worthy of attention.

Objectives: The objective of this study is to revise China's official nominal interest rate and inflation rate by representing the market nominal rate with SIBOR interest rate and by considering the GCPI of housing prices, in order to get a more real level of China's real interest rate.

Research gap filled: The marginal academic contribution of this study is considering the revision of China's nominal interest rate and official inflation rate at the same time, and using the revised indicators to measure China's real interest rate level in the past 20 years, thus providing empirical support for judging China's financial deepening level. 

Comment 4: Introduce the various section in last part.

Reply 4: Thank you for the comments, the outline of the paper is added in the last part as follow: The outline of this study mainly includes: Relevant data are processed to provide a basis for further comparison within the same time range. The proportion of housing purchase expenditure is calculated in total household consumption, and use this weight to modify China's official CPI into GCPI. SHIBOR interest rate is used to replace the nominal interest rate, and use SHIBOR interest rate to minus GCPI to obtain China's real interest rate in recent years. It is found that China's real interest rate is negative in most of recent years, which is quite different from the results calculated directly with China's official data.

Comment 5: 2.1, 2.3, 2.4, missing citation

Reply 4: Thank you for the comments, two relevant citations were added in 2.1 as follow:

[35]Hao J, He F, Liu-Chen B, et al. Price discovery and its determinants for the Chinese soybean options and futures markets[J]. Finance Research Letters, 2021, 40: 101689.

[36]Feng H, Liu Y, Wu J, et al. Financial Market Spillover Effect and Macroeconomic Shocks: Evidence from China[J]. Available at SSRN 4282634.

[37]Fang J, Lau C K M, Lu Z, et al. Bank performance in China: A Perspective from Bank efficiency, risk-taking and market competition[J]. Pacific-Basin Finance Journal, 2019, 56: 290-309.

One relevant citations were added in 2.3 as follow:

[40]Dreger C, Zhang Y. Is there a bubble in the Chinese housing market?[J]. Urban Policy and Research, 2013, 31(1): 27-39.

One relevant citations were added in 2.4 as follow:

[41]Bikker J A, Vervliet T M. Bank profitability and risk‐taking under low interest rates[J]. International Journal of Finance & Economics, 2018, 23(1): 3-18.

Comment 6: Detailed description of the data is needed.

Reply 6: Thank you for the comments, Table 1 Presentation of data values was added in 2.5 as follow:

2.5 Presentation of relevant data

 The following table 1 shows the detail data of each variables:

Table 1 Presentation of data values

Year 1999 2000 2001 2002 2003 2004 2005 2006 2007 2008 2009 2010

HS1 125300 126300 127200 128000 128800 129600 130400 131100 131800 132500 133100 125300

TP2 124200 125300 126300 127200 128000 128800 129600 130400 131100 131800 132500 133100

PP3 4868 5480 5893 6473 6903 7474 8224 8713 9361 9258 10069 10919

HPI4 0.16% 4.90% 3.54% 3.72% 5.02% 18.71% 12.58% 6.23% 16.86% -1.89% 24.69% 5.97%

CPI5 -1.40% 0.35% 0.72% -0.73% 1.13% 3.82% 1.78% 1.65% 4.82% 5.93% -0.73% 3.18%

Year 2011 2012 2013 2014 2015 2016 2017 2018 2019 2020 2021 2022

HS1 133800 134500 135400 136300 137200 138000 138800 139600 140300 140800 141100 141200

TP2 134500 135400 136300 137200 138000 138800 139600 140300 140800 141100 141200 141175

PP3 11613 12278 12850 13894 15589 17869 18322 19853 21559 21210 24100 24538

HPI4 5.67% 8.75% 7.73% 1.42% 9.10% 11.28% 5.71% 12.33% 8.58% 7.46% 4.17% -2.03%

CPI5 5.55% 2.62% 2.62% 1.92% 1.44% 2.00% 1.59% 2.07% 2.90% 2.42% 0.98% 2.00%

1.Ten thousand yuan 2.Ten thousand people 3. Yuan 4. Percent 5.Percent

Comment 7: Extend section 2.4

In order to explain the measurement more clearly, the following sentence was added in the last part of 2.4: 

2.4 Measurement of China's Real Interest Rate Based on GCPI

In order to evaluate China's real interest rate after deducting inflation from different perspectives [41], four indicators were selected: CPI, GCPI, one-year fixed deposit rate, and SIBOR. The traditional real interest rate and the real interest rate calculation based on GCPI are analyzed separately, and the results were compared. By subtracting the inflation rate (CPI and GCPI) from the nominal interest rate (SIBOR and one-year fixed deposit) , this paper shows the calculation results of China's real interest rate from different perspectives. Different from relevant studies that focus on the influencing factors of China's real estate market price (including the impact of market interest rates on housing prices) [42][43], one of the innovations of this paper is to study the changes of China's real interest rates after housing prices are included from the data level. This study updates the calculation of China's real interest rates on one hand, and also provides reference for academic research and policy formulation on the other.

Comment 7: Interest rate has been studied extensively in previous literature of housing price research such as:

https://www.mdpi.com/2071-1050/10/2/341

https://www.researchgate.net/publication/359519947_Factors_Affect_the_Housing_Prices_in_China_A_Systematic_Review_of_Papers_Indexed_in_Chinese_Science_Citation_Database

What is the new point in this research then? Please compare your research and then with previous research to highlight the research gap that it fills. One subsection is needed for interest rate research impact on housing price.

Reply 7: Thank you for the comments, we have carefully read the two papers that you have mentioned, and modified the 2.4 as follow with two more citations: 

[42]Li N, Li R Y M, Nuttapong J. Factors affect the housing prices in China: a systematic review of papers indexed in Chinese Science Citation Database[J]. Property Management, 2022 (ahead-of-print).

[43]Xu Y, Ma Y, Zhu Z, et al. Construct comprehensive indicators through a signal extraction approach for predicting housing price crises[J]. Plos one, 2022, 17(8): e0272213.

The part 2.4 is revised as described in reply 6.

Comment 8: Extend the results part and then make discussion an independent section. Discussion has to be done based on previous literature.

Reply 8: Thank you for the comments, we have added a new paragraph in the end Part 3 as follow: 

3.4 What changes have GCPI brought to China's real interest rate calculation

Both the ordinary people and relevant scholars believe that China's actual inflation level is underestimated by the official CPI [45], and the most crucial reason is that the official CPI does not include the rapid rise and high cost of housing purchase. This study is based on public statistical data and uses a clear weighting methods to synthesize the weights of residents' house purchases increase in house prices with official CPI, generating a new GCPI. In the past 24 years, if GCPI is used as the representative of inflation, whether SIBOR or one-year fixed deposit is used as the nominal interest rate, China's real interest rate in most years is negative. This result is not only in line with the perception of the ordinary people, but also reflected the the financial repression policies that still exist in China. It is worth noting that in the second decade of the 21st century, due to the gradual cooling of China's real estate market, the actual value of GCPI showed a lower trend than CPI. This result confirms the accuracy of using GCPI as an inflation measure from another perspective, as research has pointed out that there is a risk of deflation in the current Chinese economy [46].

[45]Pan L, Amin A, Zhu N, et al. Exploring the Asymmetrical Influence of Economic Growth, Oil Price, Consumer Price Index and Industrial Production on the Trade Deficit in China[J]. Sustainability, 2022, 14(23): 15534.

[46]Thiagarajan R, Im H, Khasnis A. The End of an Era[M] State Street. 2023.

Comment 9: Calculation of interest rate CPI is not needed much. There is not much academic and practical value in new calculations of CPI and interest rate. 

Reply 9: Thank you for the comments, in order to calculate the actual interest rate, it is inevitable to know the accurate inflation rate, and the calculation of the actual interest rate helps to understand the depth of China's financial market. So we added the following content in Introduction part: “The marginal academic contribution of this study is considering the revision of China's nominal interest rate and official inflation rate at the same time, and using the revised indicators to measure China's real interest rate level in the past 20 years, thus providing empirical support for judging China's financial deepening level.” 

Comment 10: All tables need citations, so as the calculations.

Reply 10: Thank you for the comments, footnotes were added for each figure.

Comment 11: A more complicated model is needed. Now is a bit too simple.

Reply 11: Thank you for the comments, The calculation method of China's CPI involves two core steps: The first step is determining the basket of goods and services that represents the consumption patterns of urban households in China. This basket includes a wide range of items, such as food, housing, transportation, healthcare, education, and other goods and services. Then each item in the basket is assigned a weight that reflects its relative importance in household expenditure. The weights are based on consumption surveys conducted periodically to capture changes in consumption patterns. Considering that China's official CPI adopts those weighted summation method, in order to ensure consistency in the calculation of GCPI, this manuscript also adopts a weighted summation method similar to CPI calculation, so that the calculations of the two indices can be integrated with each other. Of course, in the future, higher-level methods such as the dynamic factor index method can be used to further refine the calculation of inflation rates.

Comment 12: Extend conclusion, state limitations and future research direction.

Reply 12: Thank you for the comments, the last part of this manuscript is rewritten as follow:

The core idea of this study's calculation method is to assume that the expenditure corresponding to China's official CPI is the total expenditure of residents minus the expenditure on purchasing houses. Then, the weights of CPI and HPI are calculated through relevant statistical data, and the two are combined into GCPI. The foundation of this approach is still based on statistical data measurement, and in the future, improvements can be made from two aspects: firstly, improving CPI measurement through big data methods; another approach is to use higher-level methods such as dynamic factor analysis to improve the synthesis of HPI and CPI [47].

Comment 13: References [J] is not needed.

Reply 13: Thank you for the comments, the format of the references has been adjusted according to the needs of the journal.

Comment 14: Font size of the words need to be standardized.

Reply 14: Thank you for the comments, The format of the manuscript has been adjusted according to the needs of the journal. 

Review 2 

Comment 1: Authors proposed a new measure of China's real interest rate that generalizes the usual Consumer Price Index (CPI). They compares CPI with proposed Generalized CPI and provide their benefits. Then they related the Generalized CPI with Shanghai Interbank Offered Rate (SIBOR) and study its impact.

Reply 1: Thank you for the comments! 

Comment 2: I think that authors provide a new insight in the area and discuss appropriately their benefits. The new measure could be considered in future works. I could recommend the publication of this manuscript in Plos One if authors addressed the following issues:

Reply 2: Thank you for the comments! We have made thoroughly revisions to the article based on your and other reviewers' opinions, and hope to meet the publishing requirements of PLOS ONE with your help.

Comment 3: Line 16: remove "generally".

Reply 3: Thank you for the comments! The first sentence of the abstract was revised as follow: “The Chinese economy has undergone a long-term transition reform, but there is still a planned economy characteristic in the financial sector, which is financial repression. Due to the existence of financial repression, China's actual interest rate level should be lower than the Consumer Price Index (CPI).”

Comment 4: Line 17: "CPI" <-> "Consumer Price Index (CPI)".

Reply 4: Thank you for the comments! The full name of CPI has been described in the first sentence added to the abstract. 

Comment 5: Line 80: "Second" <-> "second".

Reply 5: Thank you for the comments! Revised.

Comment 6: Line 84: "2.1. Treatment of data".

Reply 5: Thank you for the comments! Revised.

Comment 7: Eqs. (1)-(3): remove these points.

Reply 7: Thank you for the comments! Revised.

Comment 8: Replace "formula" by "Equation".

Reply 8: Thank you for the comments! Each 'formula' in the manuscript have been replaced with ' equation '

Comment 9: Line 138: "can be obtained in Figure 1 from 1999-2022.".

Reply 8: Thank you for the comments! Revised.

Comment 10: Line 222: Add as further comment the use of statistical method for the study of comparisons between Generalized CPI and SIBOR, or another important variables related to resources, such as in Idrovo-Aguirre and Contreras-Reyes (2021). Reference: Idrovo-Aguirre, B.J., Contreras-Reyes, J.E. (2021). The response of housing construction to a copper price shock in Chile (2009–2020). Economies, 9(3), 98.

Reply 10: Thank you for the comments! We have carefully read the reference that you recommended, and the last part of the conclusion was revised as follow with the reference recommended:

The core idea of this study's calculation method is to assume that the expenditure corresponding to China's official CPI is the total expenditure of residents minus the expenditure on purchasing houses. Then, the weights of CPI and HPI are calculated through relevant statistical data, and the two are combined into GCPI. The foundation of this approach is still based on statistical data measurement, and in the future, improvements can be made from two aspects: firstly, improving CPI measurement through big data methods; another approach is to use higher-level methods such as dynamic factor analysis to improve the synthesis of HPI and CPI [47].

Review 3

Comment 1: First of all, the English writing of this manuscript needs to be improved.

Reply 1: Thank you for the comments! We have carefully revised the English writing according to the guild of an English professor. 

Comment 2: Secondly, it is suggested that the author add a literature review to specifically summarize the connotation and measurement methods of Real Interest Rate, and find that it is not enough to highlight the value and innovation of this study.

Reply 2: Thank you for the comments! We deeply agree that the literature review of the Real Interest Rate is not enough. So after carefully literature review, the beginning of the fourth paragraph was revised as follow with two more citations:

Financial repression often lead to negative real interest rates [19]. There are several classical methods to estimate the real interest rates [20][21]

[20]King M, Low D. Measuring the''world''real interest rate[R]. National Bureau of Economic Research, 2014.

[21]Kiley M T. The global equilibrium real interest rate: concepts, estimates, and challenges[J]. Annual Review of Financial Economics, 2020, 12: 305-326.

In order to highlight the value and innovation of this study, the sixth paragraph of the introduction was rewritten as follow:

The objective of this study is to revise China's official nominal interest rate and inflation rate by representing the market nominal rate with SIBOR interest rate and by considering the GCPI of housing prices, in order to get a more real level of China's real interest rate. The specific measures include the following two points: first, using SIBOR interest rate as the measurement standard of market interest rate (nominal interest rate); second, adopting a certain method to incorporate the rise of housing prices into the existing CPI system, thereby correcting the limitations of China's existing CPI measurement. The marginal academic contribution of this study is considering the revision of China's nominal interest rate and official inflation rate at the same time, and using the revised indicators to measure China's real interest rate level in the past 20 years, thus providing empirical support for judging China's financial deepening level.

Comment 3: Finally, China's CPI includes rents, not housing prices. Usually, rents and housing prices move in tandem. Why doesn't China's CPI include housing prices? Housing is an investment property, so it is likely that housing prices rise and price of commodities remains stable or price of commodities rises and housing prices remain stable. The authors need to clarify the underlying reasons why China's CPI excludes housing prices.

Reply 3: Thank you for the comments! We deeply agree this manuscript does not highlight the reasons why housing prices are included in CPI and the necessity of correcting CPI. Therefore, in the fifth paragraph of the introduction, the following description is made with 3 references:

In response to the problem that market interest rates cannot be fully represented by one-year fixed deposits, academia has pointed out that the Shanghai Interbank Offered Rate (SIBOR) can be used as a measurement indicator of market interest rates [29], while many studies have pointed out that house consumer price factors should be taken into account in the revision of CPI [30], since the housing prices are rising too fast and costed a lot [31]. China's CPI does indeed include changes in housing rent, but the rent-price ratio of Chinese housing is lower than the widely recognized level for longtime [32], and purchasing house in China is a necessity among residents [33]. Therefore, considering only housing rents cannot represent the level of expenditure by residents. This study referenced the CPI calculation methods of developed countries such as the United States that consider housing purchase factors [34], and attempted to incorporate the change in housing purchase expenditure prices into the CPI measurement system, in order to better characterize China's actual inflation rate.

[32]Zhai D, Shang Y, Wen H, et al. Housing price, housing rent, and rent-price ratio: Evidence from 30 cities in China[J]. Journal of Urban Planning and Development, 2018, 144(1): 04017026.

[33]Or T. Pathways to homeownership among young professionals in urban China: The role of family resources[J]. Urban Studies, 2018, 55(11): 2391-2407.

[34]André C, Gabauer D, Gupta R. Time-varying spillovers between housing sentiment and housing market in the United States[J]. Finance Research Letters, 2021, 42: 101925.

Reviewer 4

Comment 1: I have carefully studied Manuscript Study on the Measurement of China's Real Interest Rate after Including Housing Price Factor and have the following observations for the author(s):

Reply 1: Thank you for the comments! We have carefully revised the manuscript based on your observations.

Comment 2: Q1) The abstract is too short and does not provide useful details to the readers. The authors have done a great job but they have not explained their work in the Abstract. Please follow some previous papers and provide background, findings, methodology, and brief policies in this section to make it more attractive to readers

Reply 2: Thank you for the comments! We have carefully revised the Abstract part as follow:

The Chinese economy has undergone a long-term transition reform, but there is still a planned economy characteristic in the financial sector, which is financial repression. Due to the existence of financial repression, China's actual interest rate level should be lower than the Consumer Price Index (CPI). However, based on official China’s interest rates and CPI, over half of the years China’s actual interest rate remained higher than CPI by our calculation from 1999 to 2022. This is inconsistent with the financial repression that exists in China, and the main reason is the calculation methods of China's CPI. China's CPI measurement system originated from the planned economy era, which did not fully consider the rise in housing purchase prices, so the current CPI measurement system can be more realistically presented by taking the rise in housing prices into consider. The core idea of this study is to mining relevant official statistical data and calculate the proportion of Chinese residents' expenditure on purchasing houses to their total expenditure. By taking the proportion of house purchases as the weight of house price factor, and taking the proportion of other consumption as the weight of official CPI, the Generalized CPI (GCPI) is formulated. The GCPI is then compared with market interest rates to determine the actual interest rate situation in China over the past 20 years. This study has found that if GCPI is used as a measure, China's real interest rates have been negative for most years since 1999. Chinese residents have suffered the negative effects of financial repression over the past 20 years, and their property income cannot keep up with the actual losses caused by inflation. Therefore, it is believed that China's CPI calculation method should be adjusted to take into account the rise in housing prices, so China's actual inflation level could be more accurately reflected. In view of the above, deepening interest rate marketization reform and expand channels for financial investment are the future development goals of China's financial system.

Comment 3: Q2 The authors need to more empirical tests:

Reply 3: Thank you for the comments! The current research mainly uses the method of calculating the GCPI (Generalized CPI) including housing prices through the proportion of housing purchase expenses. Other higher-order methods will be implemented in future research, and a section of future research has been added at the end of this manuscript:

The core idea of this study's calculation method is to assume that the expenditure corresponding to China's official CPI is the total expenditure of residents minus the expenditure on purchasing houses. Then, the weights of CPI and HPI are calculated through relevant statistical data, and the two are combined into GCPI. The foundation of this approach is still based on statistical data measurement, and in the future, improvements can be made from two aspects: firstly, improving CPI measurement through big data methods; another approach is to use higher-level methods such as dynamic factor analysis to improve the synthesis of HPI and CPI 

[47] Idrovo-Aguirre B J, Contreras-Reyes J E. The response of housing construction to a copper price shock in Chile (2009–2020)[J]. Economies, 2021, 9(3): 98.

Comment 4: Q3 Please, provide those theoretical reasoning that underpin the direction of the authors research investigations and the econometric model used in the study

Reply 4: Thank you for the comments! According to your comment, the sixth paragraph of this manuscript was rewritten as follow: 

The objective of this study is to revise China's official nominal interest rate and inflation rate by representing the market nominal rate with SIBOR interest rate and by considering the GCPI of housing prices, in order to get a more real level of China's real interest rate. The specific measures include the following two points: first, using SIBOR interest rate as the measurement standard of market interest rate (nominal interest rate); second, adopting a certain method to incorporate the rise of housing prices into the existing CPI system, thereby correcting the limitations of China's existing CPI measurement. The marginal academic contribution of this study is considering the revision of China's nominal interest rate and official inflation rate at the same time, and using the revised indicators to measure China's real interest rate level in the past 20 years, thus providing empirical support for judging China's financial deepening level.

Comment 5: Q4 The interpretations of results should be modified to have scientific reasoning. The author(s) need to provide the interpretations of results for each variable analysed in terms of real time happening of events. The interpretations should as well be expanded to provide the implications of both positive and negatives findings to the individual case variable under focus and to also provide the how and why the findings in the result went the way it did with respect to the case study area. Each findings discovered by the study should be compared with past studies and draw lesson and head-ways from the perspective of the study’s objectives and the relevant theories provided in the study.

Reply 5: Thank you for the comments! The last part of results was added as follow: 

3.4 What changes have GCPI brought to China's real interest rate calculation

Both the ordinary people and relevant scholars believe that China's actual inflation level is underestimated by the official CPI [45], and the most crucial reason is that the official CPI does not include the rapid rise and high cost of housing purchase. This study is based on public statistical data and uses a clear weighting methods to synthesize the weights of residents' house purchases increase in house prices with official CPI, generating a new GCPI. In the past 24 years, if GCPI is used as the representative of inflation, whether SIBOR or one-year fixed deposit is used as the nominal interest rate, China's real interest rate in most years is negative. This result is not only in line with the perception of the ordinary people, but also reflected the the financial repression policies that still exist in China. It is worth noting that in the second decade of the 21st century, due to the gradual cooling of China's real estate market, the actual value of GCPI showed a lower trend than CPI. This result confirms the accuracy of using GCPI as an inflation measure from another perspective, as research has pointed out that there is a risk of deflation in the current Chinese economy [46].

[46]Thiagarajan R, Im H, Khasnis A. The End of an Era State Street. 2023.

Comment 6: Q5 Some recent literature relating to interest rate must be added. Some suggestions are as follows:

Xu, Y., Ma, Y., Zhu, Z., Li, J., & Lu, T. (2022). Construct comprehensive indicators through a signal extraction approach for predicting housing price crises. Plos one, 17(8), e0272213.

Samour, A., Isiksal, A. Z., & Gunsel Resatoglu, N. (2020). The impact of external sovereign debt and the transmission effect of the US interest rate on Turkey’s equity market. The Journal of International Trade & Economic Development, 29(3), 319-333.

Reply 6: Thank you for the comments! We have carefully read the literature that you recommend, and added them in [43] and [44].

Reviewer 5

Comment 1: Manuscript is not well written in English.

Reply 1: Thank you for the comments! We have carefully revised the English writing according to the guild of an English professor.

Comment 2: Title can be improved ( title not catchy/ convincing at all). title is misleading, this is not a study in general, this is more like new calculation on CPI

Reply 2: Thank you for the comments! The title was revised into : “Study on the China's Real Interest Rate after Including Housing Price Factor into CPI”

Comment 3: The first 2 sentences in the abstract does not attract the reader to read the whole paper. Can be improved further.

Reply 3: Thank you for the comments! We have carefully revised the Abstract part as follow:

The Chinese economy has undergone a long-term transition reform, but there is still a planned economy characteristic in the financial sector, which is financial repression. Due to the existence of financial repression, China's actual interest rate level should be lower than the Consumer Price Index (CPI). However, based on official China’s interest rates and CPI, over half of the years China’s actual interest rate remained higher than CPI by our calculation from 1999 to 2022. This is inconsistent with the financial repression that exists in China, and the main reason is the calculation methods of China's CPI. China's CPI measurement system originated from the planned economy era, which did not fully consider the rise in housing purchase prices, so the current CPI measurement system can be more realistically presented by taking the rise in housing prices into consider. The core idea of this study is to mining relevant official statistical data and calculate the proportion of Chinese residents' expenditure on purchasing houses to their total expenditure. By taking the proportion of house purchases as the weight of house price factor, and taking the proportion of other consumption as the weight of official CPI, the Generalized CPI (GCPI) is formulated. The GCPI is then compared with market interest rates to determine the actual interest rate situation in China over the past 20 years. This study has found that if GCPI is used as a measure, China's real interest rates have been negative for most years since 1999. Chinese residents have suffered the negative effects of financial repression over the past 20 years, and their property income cannot keep up with the actual losses caused by inflation. Therefore, it is believed that China's CPI calculation method should be adjusted to take into account the rise in housing prices, so China's actual inflation level could be more accurately reflected. In view of the above, deepening interest rate marketization reform and expand channels for financial investment are the future development goals of China's financial system.

Comment 4: No literature review section/ not enough literature to support the objective of the study.

Reply 4: Thank you for the comments! We have carefully read a large number of references, and based on the reviewers' comments, increased the number of existing references to 47. 

Comment 5: No analysis conducted, it is just calculation of new CPI

Reply 5: Thank you for the comments! According to your comment, the “2.4 Measurement of China's Real Interest Rate Based on Generalized CPI” part was rewritten as follow: 

2.4 Measurement of China's Real Interest Rate Based on GCPI

In order to evaluate China's real interest rate after deducting inflation from different perspectives [41], four indicators were selected: CPI, GCPI, one-year fixed deposit rate, and SIBOR. The traditional real interest rate and the real interest rate calculation based on GCPI are analyzed separately, and the results were compared. By subtracting the inflation rate (CPI and GCPI) from the nominal interest rate (SIBOR and one-year fixed deposit) , this paper shows the calculation results of China's real interest rate from different perspectives. Different from relevant studies that focus on the influencing factors of China's real estate market price (including the impact of market interest rates on housing prices) [42][43], one of the innovations of this paper is to study the changes of China's real interest rates after housing prices are included from the data level. This study updates the calculation of China's real interest rates on one hand, and also provides reference for academic research and policy formulation on the other.

Thank you again for the work of editors and reviewers.

 Sincerely Yours,

First author on behalf of all authors: Ding Shiting

---

## [Decision Letter · Decision Letter 1]

1 Aug 2023

Study on the China's Real Interest Rate after Including Housing Price Factor into CPI

PONE-D-23-10280R1

Dear Dr. Jingdong Luan,

We’re pleased to inform you that your manuscript has been judged scientifically suitable for publication and will be formally accepted for publication once it meets all outstanding technical requirements.

Kind regards,

Ricky Chee Jiun Chia

Academic Editor

PLOS ONE

Additional Editor Comments (optional):

Reviewers' comments:

Reviewer's Responses to Questions

**Comments to the Author**

1. If the authors have adequately addressed your comments raised in a previous round of review and you feel that this manuscript is now acceptable for publication, you may indicate that here to bypass the “Comments to the Author” section, enter your conflict of interest statement in the “Confidential to Editor” section, and submit your "Accept" recommendation.

Reviewer #2: All comments have been addressed

Reviewer #6: All comments have been addressed

2. Is the manuscript technically sound, and do the data support the conclusions?

Reviewer #2: Yes

Reviewer #6: (No Response)

3. Has the statistical analysis been performed appropriately and rigorously? 

Reviewer #2: Yes

Reviewer #6: Yes

4. Have the authors made all data underlying the findings in their manuscript fully available?

Reviewer #2: Yes

Reviewer #6: Yes

5. Is the manuscript presented in an intelligible fashion and written in standard English?

Reviewer #2: Yes

Reviewer #6: No

6. Review Comments to the Author

Reviewer #2: Authors addressed in well form all my previous comments and suggestions. I don't have further comments.

Reviewer #6: i have checked the Study on the China's Real Interest Rate after Including Housing Price Factor into CPI

the authors revised the paper based on my comments ,

my decision is accepted

7. PLOS authors have the option to publish the peer review history of their article (what does this mean?). If published, this will include your full peer review and any attached files.

Reviewer #2: No

Reviewer #6: No

---

## [Editor Report · Acceptance letter]

4 Aug 2023

PONE-D-23-10280R1 

Study on the China's Real Interest Rate after Including Housing Price Factor into CPI 

Dear Dr. Luan:

I'm pleased to inform you that your manuscript has been deemed suitable for publication in PLOS ONE. Congratulations! Your manuscript is now with our production department. 

Kind regards, 

on behalf of

Dr. Ricky Chee Jiun Chia 

Academic Editor

PLOS ONE